# ADMP-GNN: Adaptive Depth Message Passing GNN

## Abstract

Graph Neural Networks (GNNs) have proven to be highly effective in various graph representation learning tasks. A key characteristic is that GNNs apply a fixed number of message-passing steps to all nodes in the graph, regardless of the varying computational needs and characteristics of each node. Through empirical analysis of real-world data, we show that the optimal number of message-passing layers differs for nodes with different characteristics. This insight is further validated with experiments on synthetic datasets. To address this, we propose Adaptive Depth Message Passing GNN (ADMP-GNN), a novel framework that dynamically adjusts the number of message-passing layers for each node, leading to enhanced performance. This approach is applicable to any model that follows the message-passing scheme. We evaluate ADMP-GNN on the node classification task and observe performance improvements over a wide range of GNNs.

## 1 Introduction

A plethora of structured data comes in the form of graphs (Bornholdt & Schuster, 2001; Cao et al., 2020); this has driven the need to develop neural network models that can effectively process and analyze graph-structured data, known as Graph Neural Networks (GNNs). GNNs recently gathered increasing attention following their successes in learning complex node and graph representations, showcasing impressive success in various applications (Corso et al., 2022; Rampášek et al., 2022). Many GNNs are instances of Message Passing Neural Networks (MPNNs) (Gilmer et al., 2017a) such as Graph Isomorphism Networks (GIN) (Xu et al., 2019) and Graph Convolutional Networks (GCN) (Kipf & Welling, 2017). A common characteristic of GNNs is that they typically employ a fixed number of message-passing steps for all nodes, determined by the number of layers in the GNN (Gilmer et al., 2017b). This static approach raises an intriguing question: *Should the number of message-passing steps be adapted individually for each node to better capture their unique characteristics and computational needs?*

Determining the optimal number of message-passing layers for each node in a Graph Neural Network (GNN) presents a significant challenge due to the intricate and diverse nature of graph structures, node features, and learning tasks. While deeper GNNs are capable of capturing long-range dependencies (Liu et al., 2021), they can also encounter issues like oversmoothing, where nodes become indistinguishably similar (Luan et al., 2022). This underscores the critical importance of selecting the appropriate number of layers for a GNN to effectively capture the necessary graph information. In dense graphs, where information can propagate quickly, even shallow GNNs can effectively capture local information (Zeng et al., 2020). Conversely, sparse graphs, particularly those with isolated nodes or limited connectivity, may require additional layers to facilitate effective information sharing (Zhang et al., 2021; Zhao & Akoglu, 2020). An even more compelling idea is to adjust the GNN depth for each node based on its local complexity and structural properties. This adaptive approach could be especially beneficial for graphs with varied local structures, ensuring that each node is processed according to its unique requirements.

Dynamic Neural Networks, also known as Adaptive Neural Networks, represent a class of models that possess the ability to adjust their architecture or parameters depending on the input. Dynamic Neural Networks have gained significant popularity, especially in the field of computer vision. This adaptability enables them to achieve improved performance metrics such as accuracy, computational efficiency, and robustness. The adaptation includes, for example, the number of layers and skip

connections (Li et al., 2017; Huang et al., 2016; Sabour et al., 2017). However, applying these adaptations to graph learning tasks presents unique challenges. While Dynamic Neural Networks excel in structured and homogeneous data environments like computer vision, graph data involves overcoming complexities related to the inherent complex structure of graphs. Moreover, for node classification tasks, input samples are interconnected through graph edges, necessitating specialized techniques to dependencies.

In this work, we focus on the task of node classification by proposing ADMP-GNN, a novel approach that dynamically adapts the number of layers for each node within a GNN. Our main contributions are as follows:

1. **Node-Specific Depth Analysis in Graph Neural Networks.** We demonstrate through empirical analysis that different nodes within the same graph may require varying numbers of message-passing steps to accurately predict their labels. This finding underscores the importance of node-specific depth in GNNs.

2. **Adaptive Message-Passing Layer Integration.** We present ADMP-GNN, a novel approach that enables any GNN to make predictions for each node at every layer. Training the GNN to predict labels across all layers is a multi-task setting, which often suffers from gradient conflicts leading to suboptimal performance. To address this, we propose a sequential training methodology where layers are progressively trained and their gradients are subsequently frozen, thereby mitigating conflicts and improving overall performance.

3. **Adaptive Layer Policy Learning for Node Classification.** We propose a heuristic method to learn a layer selection policy using a set of validation nodes. This policy is then applied to select the optimal layer for predicting the labels of test nodes, ensuring that each node exits the GNN at the most appropriate layer for its specific classification task.

4. **Model-Agnostic Flexibility.** Our approach is model-agnostic and can be integrated with any GNN architecture that employs a message-passing scheme. This flexibility enhances the GNN's performance on node classification tasks, providing a significant improvement over traditional fixed-layer approaches.

## 2 RELATED WORK

### 2.1 GRAPH NEURAL NETWORKS

Graph Neural Networks (GNNs) are a class of deep learning methods that operate on graph data. As most neural networks, GNNs are formed by stacking many layers. Each layer, $\ell$, is responsible for updating the node representations $\{h_u^{(\ell)}, u \in \mathcal{V}\}_{0 \leq \ell \leq L}$, relying on the graph structure and the output from the previous layer, $\{h_u^{(\ell-1)}, u \in \mathcal{V}\}$. The goal of a GNN layer $\ell$ is to update node representations relying mainly on the structure of the graph $(V, E)$ and the output of the previous layer $H^{(\ell-1)}$. Conventionally, the nodes features are used as input of the first layer $\{h_u^{(0)}, u \in \mathcal{V}\} = [x_v]_{v \in \mathcal{V}} \in \mathbb{R}^{N \times d}$, where $N$ is the number of nodes and $d$ is the features dimension. A basic GNN layer is based on the message passing mechanism and consists of two components: *(i) Aggregate Layer* $\psi$ that applies for each node $v$, a permutation invariant function to its neighbors, denoted by $\mathcal{N}(v)$ to generate the aggregated node feature; *(ii) Update Layer* $\phi$ that combines the aggregated node feature $m_v^{(\ell)}$ with the previous hidden vector $h_v^{(\ell-1)}$, and generate a new representation $h_v^{(\ell)}$ of the same node $v$:

$$m_v^{(\ell)} = \psi^{(\ell)}(\{h_u^{(\ell-1)}, u \in \mathcal{N}(v)\}),$$
$$h_v^{(\ell)} = \phi^{(\ell)}(h_v^{(\ell-1)}, m_v^{(\ell)}).$$

Depending on the task, an additional readout or pooling function can be added after the last layer to aggregate the representation of nodes.

$$h_G = \texttt{READOUT}(H^{(\ell)}).$$

### 2.2 DYNAMIC-DEPTH NEURAL NETWORKS

Dynamic neural networks have emerged as a focal point in the realm of deep learning. Unlike static models with fixed computational graphs and parameters during inference, dynamic networks

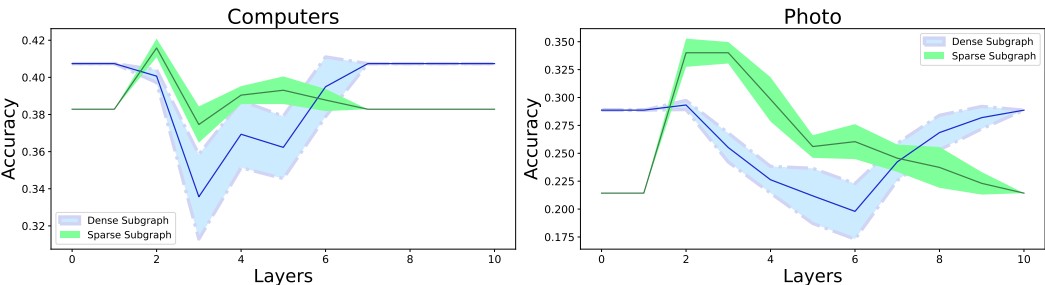

Figure 1: Effect of GCN Depth on Sparse and Dense Subgraphs: The figure shows the performance of GCN models when varying layer depths, and comparing its effectiveness on both sparse and dense subgraphs.

adapt their structures or parameters based on varying inputs. This dynamic flexibility gives models significant benefits, such as improved accuracy, enhanced computational efficiency, and superior adaptability (Wang et al., 2020; Zhou et al., 2020). A popular type of dynamic neural networks includes those that dynamically adjust network depth based on each input. For instance in natural language processing, some adaptive large language models employ adaptive depth to optimize both inference speed and computational memory usage of the Transformer architecture (Elbayad et al., 2020; Schuster et al., 2022; Vaswani et al., 2017). In the field of computer vision, there are studies that dynamically generate filters conditioned on each input, enhancing flexibility without significantly increasing the number of model parameters (Jia et al., 2016).

To the best of our knowledge, in the field of GNNs, there has been no prior work proposing adaptive depth for each node. However, several studies have focused on combining all GNN layers. These works typically aim to adapt GNN architectures for heterogeneous graphs (Chien et al., 2020) and leverage information from higher-order neighbors (Xu et al., 2018a). Additionally, other related approaches here concern residual connections to mitigate issues like oversmoothing (Chen et al., 2020). While combining GNN layers can be viewed as a form of depth-adaptive strategy, where the final node representation is guided by the optimal intermediate hidden states, this approach remains static because the same inference policy is applied uniformly across all nodes and learned layer aggregators stay fixed after training.

## 3 ADMP-GNN: ADAPTIVE DEPTH MESSAGE-PASSING BASED GNN

In this section, we first present an empirical analysis highlighting the necessity for node-specific depth in GNNs. Then, we introduce our *Adaptive Depth Message Passing-based GNN (ADMP-GNN)*. Our study includes experiments on both synthetic and real-world datasets to illustrate the importance and potential benefits of this methodology.

### 3.1 NODE-SPECIFIC DEPTH ANALYSIS IN GRAPH NEURAL NETWORKS

**Analysis on Synthetic Graphs.** The goal of this analysis is to motivate the need to use a varying number of message-passing steps based on the specific characteristics of individual nodes. As discussed in the introduction, this approach becomes especially interesting in hybrid graphs where nodes exhibit diverse properties, such as local structures and node features. In this experiment, we focus on analyzing the impact of the number of message-passing layers on node with varied local neighborhood sparsity. To do so, we construct a graph by merging two subgraphs extracted from a real-world dataset, such as Computers and Photo (Shchur et al., 2018). Both subgraphs contain the same number of nodes, exhibit nearly identical homophily, and have equally distributed node labels. The main difference between these subgraphs lies in their structure as one subgraph is sparse, while the other is dense. Consequently, nodes within each subgraph share similar structural characteristics. In Appendix A, we give the construction details of these synthetic datasets, and we visualize the adjacency matrix of these synthetic graphs, with additional details provided in the appendix, c.f. Figure 3.

We trained a $L$ different GCN models, with a varying number of layers $\ell \in [\![0, L]\!]$, where $L$ represents the maximum depth. For this experiment, we set $L = 10$. Although each GCN was trained on the entire synthetic graph, which is a combination of the sparse and dense subgraphs, we evaluated the performance of each model separately on the individual subgraphs. This allows us to assess the impact of GNN depth on different types of local subgraph structures. The results of this analysis are presented in Figure 1. As observed, in dense subgraphs, the test accuracy decreases at a faster rate, while in sparse subgraphs, the drop in accuracy occurs later, typically around layers 2 or 3. Moreover, the optimal number of layers differs between sparse and dense subgraphs. For instance, in the Computers dataset, the highest accuracy is achieved at layer 2 for the sparse subgraph, while for the dense subgraph, the optimal performance is reached at layer 0. Additionally, in the Photo dataset, we observe a distinct behavior starting from layer 6, where the impact of GNN depth diverges between sparse and dense subgraphs. This highlights the need to adapt the number of layers per node based on its characteristics.

**Analysis on Real Word Graphs.** Given a maximum GNN depth $L$, we should train $L + 1$ different GNNs, each with a distinct number of layers $\ell$, where $\ell$ ranges from 0 and $L$. Subsequently, a policy must be established to determine the optimal GNN with the appropriate number of layers for each individual node. However, training $L + 1$ GNNs separately can be computationally expensive. A more efficient approach involves designing a single GNN with $L + 1$ layers that provides predictions at each intermediate layer. To ensure that this new configuration is equivalent to the previous approach (i.e., training $L + 1$ GNNs separately), the computational graph responsible for making predictions at layer $\ell$ must be identical to that of a GNN with $\ell$ layers. Furthermore, the classification performance at layer $\ell$ should yield results comparable to those of a conventional GNN with $\ell$ layers. In what follows, we propose ADMP-GNN, an extension of message passing neural networks that respect the aforementioned challenges.

## 3.2 Adaptive Message-Passing Layer Integration

We introduce ADMP-GNN, an adaptation of a Message Passing Neural Network, with a maximum depth of $L$ layers. The goal is to ensure that the computational graph and the performance of ADMP-GNN at a certain layer matches that of traditional GNNs when trained and tested on the same number of layers. To achieve this, we incorporate an additional *Update* function, denoted as $\phi_{\mathbf{Ex}}^{(\ell)}$, to directly predict node labels at a given layer $\ell$ (**Ex** stands for 'Exit'). The function $\phi_{\mathbf{Ex}}^{(\ell)}$ is defined as follows:

$$p_v^{(\ell)} = \phi_{\mathbf{Ex}}^{(\ell)}\left(h_v^{(\ell-1)}, m_v^{(\ell)}\right) = \text{Softmax}\left(\widetilde{W}^{(\ell)} m_v^{(\ell)}\right),$$

where $\widetilde{W}^{(\ell)} \in \mathbb{R}^{d^{(\ell)} \times c}$ is a learnable weight matrix, $d^{(\ell)}$ is the dimension of the hidden representation at the $\ell$-th layer, and $c$ is the number of classes. To obtain predictions at a deeper layer $\ell' \geq \ell$, we continue the message passing using another *Update* function $\phi_{\mathbf{Ct}}^{(\ell)}$ (**Ct** stands for 'Continuation'):

$$p_v^{(\ell)} = \phi_{\mathbf{Ex}}^{(\ell)}\left(h_v^{(\ell-1)}, m_v^{(\ell)}\right),$$

$$h_v^{(\ell+1)} = \phi_{\mathbf{Ct}}^{(\ell)}(h_v^{(\ell-1)}, m_v^{(\ell)}).$$

In Figure 2, we illustrate the architecture of the proposed ADMP-GNN. For $\ell = 0$, we directly use the *Exit Update* function on the node features, i.e., $m_v^{(0)} = x_v$,

$$\forall v \in \mathcal{V}, \quad p_v^{(0)} = \phi_{\mathbf{Ex}}^{(0)}\left(m_v^{(0)}\right) = \text{Softmax}\left(\widetilde{W}^{(0)} x_v\right).$$

## 3.3 Training Scheme of ADMP-GNN

Our next objective is to train ADMP-GCN to predict node labels across all layers $\ell \in \{0, \dots, L\}$ simultaneously. For each layer $\ell$, we denote by $\theta_\ell$ the weights of the function $\psi^{(\ell)} \circ \phi_{\mathbf{Ct}}^{(\ell)}(\cdot)$. We explored two different strategies.

**Aggregate Loss Minimization (ALM).** The straightforward approach aims to optimize the sum of losses at each layer, formulated as follows,

$$\arg\min_{\theta} \mathbb{E}_{v \in \mathcal{V}}\left[S_L(v)\right] := \arg\min_{\theta_0, \dots, \theta_L} \mathbb{E}_{v \in \mathcal{V}}\left[\sum_{\ell=0}^{L} \mathcal{L}\left(p_v^{(\ell)}(m_v^{(0)}, \theta_0, \dots, \theta_\ell), y_v\right)\right], \quad (1)$$

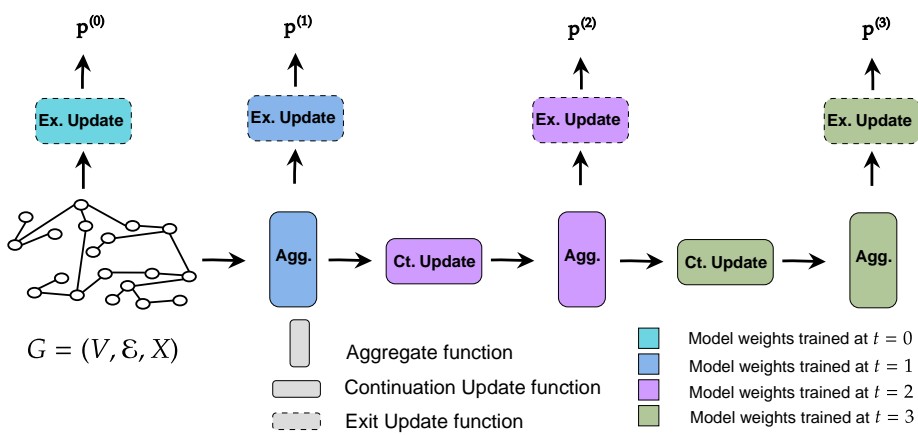

Figure 2: Illustration of ADMP-GNN, when the maximum GNN depth is $L = 3$.

where $p_v^{(\ell)}(m_v^{(0)}, \theta_0, \ldots, \theta_\ell) = \phi_{\mathbf{Ex}}^{(\ell)}(m_v^{(\ell)})$ is the prediction for the node $v$ at the layer $\ell$, and $\mathcal{L}$ is the Cross Entropy Loss. This approach may encounter gradient conflicts, particularly for early layers involved in both computation and back-propagation across upper layers.

**Sequential Training (ST).** We have studied an alternative training setup where we progressively train one GNN layer at a time, subsequently freezing each layer after training. More formally, the problem in (1) can be tackled using dynamic programming as follows:

$$\forall v \in \mathcal{V}, \quad S_{\ell+1}(v) = \mathcal{L}\left(p_v^{(\ell)}(m_v^{(0)}, \theta_0^\star, \ldots, \theta_\ell^\star, \theta_{\ell+1}), y_v\right) + S_\ell(v)$$

$$\theta_\ell^\star = \arg\min_{\theta_\ell} \mathbb{E}_{v \in V}\left[S_\ell(v)\right],$$

where $\forall v \in \mathcal{V},\ S_0(v) = \mathcal{L}(\phi_{\mathbf{Ex}}^{(0)}, y_v)$. For each intermediate layer $0 \leq \ell \leq L - 1$, by training this layer on the node classification task, we obtain high-quality node representations $[h_v^{(\ell)}]_{v \in V}$. These representations are directly employed for predictions and serve as a robust foundation for the label predictions of the subsequent layer $\ell + 1$. Algorithm 1 offers a summary of the approach.

To identify the optimal multi-task training configuration, we evaluate how much of a performance drop we lose at each layer compared to the single task setting in GNN. The comparative analysis of the three strategies is detailed in Tables 1,8,11, and 12. Our findings indicate that ADMP-GNN ST outperforms ADMP-GNN ALM. Notably, the performance of ADMP-GNN ST is comparable to, or even exceeds, that of GNN when trained under the single-task setting. Furthermore, ADMP-GNN ST exhibits a smaller standard deviation, suggesting more consistent performance.

**Time Complexity.** The training setup ST, where we sequentially train the deep ADMP-GNN, incurs relatively higher time costs due to the need for $L + 1$ training iterations. However, in each iteration, backpropagation is performed on a limited number of parameters, approximately equivalent to those in a single message passing layer. Consequently, only a small number of epochs are required for each training iteration. We report the training time of each approach in Table 5 in Appendix B.

### 3.4 ORACLE ACCURACY: EMPIRICAL JUSTIFICATION FOR NODE-SPECIFIC DEPTH IN NEURAL NETWORKS

We define *Oracle Accuracy* as the maximum test accuracy achievable by allowing a node to select the prediction from any layer. This is formally expressed as:

$$\text{Acc}_{\text{oracle}} = \frac{1}{|\mathcal{V}_{\text{test}}|} \sum_{v \in \mathcal{V}_{\text{test}}} \mathbb{1}\left\{y_v \in \bigcup_{l=0}^{L}\{\hat{y}_v^{(\ell)}\}\right\},$$

---

**Algorithm 1** Sequential Training of ADMP-GNN (ADMP-GNN ST)

---

**Inputs:** Graph $G = (\mathcal{V}, \mathcal{E})$ with node features, number of layers $L$, node classification loss function $\mathcal{L}$,

**foreach** $t \in \{0, \dots, L\}$ **do**

    **if** $t = 0$ **then**

        1. Set $h_v^{(0)} \leftarrow m_v^{(0)} = x_v$ for all $v \in \mathcal{V}$.

        2. Compute predictions at layer $\ell = 0$, i.e. $\forall v \in \mathcal{V}, \quad p_v^{(0)} = \phi_{\mathbf{Ex}}^{(0)}(h_v^{(0)})$.

        3. Train the weights of $\phi_{\mathbf{Ex}}^{(0)}$ to minimize the objective $\mathcal{L}(p_v^{(0)})$.

        4. Freeze the gradients of $\phi_{\mathbf{Ex}}^{(0)}$.

    **else**

        1. Use the *continuation* function $\phi_{\mathbf{Ct}}^{(t-1)}$ to update node representations

$$\forall v \in \mathcal{V}, \quad \tilde{h}_v^{(t-1)} = \phi_{\mathbf{Ct}}^{(t-1)}(h_v^{(t-1)}).$$

        2. Aggregate the information for neighbor nodes

$$\forall v \in \mathcal{V}, \quad m_v^{(t)} = \psi^{(t)}(\{\tilde{h}_u^{(t-1)}, u \in \mathcal{N}(v)\}).$$

        3. Compute predictions at layer $\ell = t$, i.e. $\forall v \in \mathcal{V}, \quad p_v^{(t)} = \phi_{\mathbf{Ct}}^{(t-1)}(\tilde{h}_v^{(t-1)}, m_v^{(t)})$.

        4. Train the weights of $\phi_{\mathbf{Ct}}^{(t-1)}, \psi^{(t)}$, and $\phi_{\mathbf{Ex}}^{(t)}$ to minimize the objective $\mathcal{L}(p_v^{(t)})$.

        5. Freeze the gradients of $\phi_{\mathbf{Ct}}^{(t-1)}, \psi^{(t)}$, and $\phi_{\mathbf{Ex}}^{(t)}$.

**end foreach**

---

Table 1: Comparison of three GCN training settings: *GCN* trained separately for each layer; *ADMP-GCN ALM* trained across all layers simultaneously using backpropagation on the summed loss; *ADMP-GCN ST* trained using Dynamic Programming. **(\*)** denotes single-task training.

| #layers | model | Dataset | | | | | |
|---|---|---|---|---|---|---|---|
| | | Cora | CiteSeer | CS | PubMed | Genius | ogbn-arxiv |
| 0 | GCN$^{(*)}$ | 56.38 (0.04) | 57.18 (0.12) | 88.04 (0.49) | 72.50 (0.09) | **80.82 (1.00)** | 48.88 (0.06) |
| | ADMP-GCN *ALM* | 56.96 (0.20) | 58.44 (0.21) | 87.06 (1.06) | 72.11 (0.18) | 80.03 (0.37) | 36.50 (0.12) |
| | ADMP-GCN *ST* | 56.38 (0.06) | 57.17 (0.09) | 87.27 (1.29) | 72.48 (0.14) | 80.17 (0.79) | 48.86 (0.03) |
| 1 | GCN$^{(*)}$ | 76.90 (0.14) | 69.68 (0.06) | 91.74 (0.80) | 76.63 (0.13) | 80.23 (0.37) | 55.21 (0.50) |
| | ADMP-GCN *ALM* | 75.67 (0.18) | 70.12 (0.04) | 90.55 (0.74) | 73.74 (0.16) | 80.13 (0.29) | 39.54 (1.44) |
| | ADMP-GCN *ST* | 76.90 (0.00) | 69.70 (0.00) | 90.89 (0.81) | 76.60 (0.00) | 79.93 (0.00) | 55.15 (0.00) |
| 2 | GCN$^{(*)}$ | 81.06 (0.50) | 71.05 (0.48) | **91.67 (0.94)** | **79.46 (0.31)** | 79.88 (0.51) | 66.92 (0.67) |
| | ADMP-GCN *ALM* | 70.82 (4.05) | 60.95 (2.41) | 31.2 (12.85) | 75.10 (2.14) | 79.64 (0.60) | 55.25 (0.92) |
| | ADMP-GCN *ST* | 80.73 (0.33) | **71.33 (0.40)** | 91.49 (0.66) | 79.02 (0.21) | 80.06 (0.11) | 66.51 (0.65) |
| 3 | GCN$^{(*)}$ | 79.14 (1.58) | 66.33 (1.35) | 89.80 (0.87) | 78.50 (0.68) | 80.00 (0.04) | 67.33 (0.55) |
| | ADMP-GCN *ALM* | 68.64 (5.14) | 51.30 (6.74) | 47.82 (12.98) | 74.17 (1.98) | 80.04 (0.10) | 56.22 (0.50) |
| | ADMP-GCN *ST* | 80.21 (0.52) | 70.08 (0.90) | 89.83 (1.02) | 78.25 (0.51) | 79.93 (0.00) | 68.31 (0.48) |
| 4 | GCN$^{(*)}$ | 75.96 (1.93) | 60.33 (2.38) | 78.90 (22.33) | 76.59 (0.98) | 80.01 (0.04) | 65.49 (0.99) |
| | ADMP-GCN *ALM* | 67.88 (6.05) | 49.29 (7.87) | 52.76 (11.79) | 73.40 (1.95) | 80.04 (0.10) | 56.43 (0.31) |
| | ADMP-GCN *ST* | 81.05 (0.49) | 67.99 (0.74) | 88.86 (0.70) | 75.27 (1.02) | 79.93 (0.00) | **69.29 (0.74)** |
| 5 | GCN$^{(*)}$ | 70.09 (4.01) | 57.40 (3.43) | 77.96 (15.24) | 74.32 (3.66) | 80.01 (0.04) | 63.04 (1.33) |
| | ADMP-GCN *ALM* | 67.68 (7.04) | 50.14 (7.65) | 54.53 (10.61) | 73.27 (1.51) | 80.04 (0.10) | 56.66 (0.31) |
| | ADMP-GCN *ST* | **81.22 (0.36)** | 67.42 (0.75) | 86.65 (1.52) | 75.08 (0.94) | 79.93 (0.00) | 68.92 (1.00) |

where $\mathcal{V}_{\text{test}}$ is the set of test nodes, $(\hat{y}_v^{(\ell)})_{l \leq L}$ represents the predictions for node $v$ at each layer. The oracle accuracy is greater than the accuracy at each single layer (i.e., Accuacies of ADMP-GCN PT in Table 1); taking the predictions of all nodes at a specific layer is a sub-optimal choice of the exits.

Based on the findings presented in Tables 2, 2 it is evident that the *oracle accuracy* surpasses the highest accuracies achieved by both GCN and ADMP-GCN PT. This empirical evidence strongly suggests that in an optimal configuration, employing a distinct exit layer for each node is advantageous.

## 3.5 GENERALIZATION TO TEST NODES

In this section, we introduce a heuristic approach to predict the optimal exit layer for test nodes based on the assumption that nodes exhibiting *structural similarity* should share the same exit layer. The notion of *structural similarity* can be assessed using various metrics. In our work, we define the

Table 2: The first two rows of the table present the highest accuracy ($\pm$ standard deviation) for GCN and ADMP-GCN ST, with the corresponding layer where this accuracy is achieved indicated in brackets [.]. The final row reports the *Oracle accuracy* for ADMP-GCN ST.

| model | Dataset | | | | | |
|---|---|---|---|---|---|---|
| | Cora | CiteSeer | CS | PubMed | Genius | ogbn-arxiv |
| GCN | 81.06 (0.50) [2] | 71.05 (0.48) [2] | 91.67 (0.94) [2] | 79.46 (0.31) [2] | 80.82 (1.00) [0] | 67.33 (0.55) [3] |
| ADMP-GCN *ST* | 81.22 (0.36) [5] | 71.33 (0.40) [2] | 91.49 (0.66) [2] | 79.02 (0.21) [2] | 80.17 (0.79) [0] | 69.29 (0.74) [4] |
| ADMP-GCN *ST* - Oracle | **89.43 (0.19)** | **81.96 (0.49)** | **97.24 (0.52)** | **90.13 (0.36)** | **85.97 (7.59)** | **79.64 (0.27)** |

structural similarity based on node centrality metrics, i.e., nodes are considered structurally similar if they exhibit closely aligned centrality values within the graph. We measured node centralities using both local metrics like degree and global metrics such as $k$-core (Malliaros et al., 2020), PageRank scores (Brin & Page, 1998), and Walk Count indicating the number of walks of length 2 starting from each node, which are detailed below.

**$k$-core.** We use the $k$-core decomposition of a graph, which involves iteratively removing nodes with a degree less than $k$ until no such nodes remain (Malliaros et al., 2020).

**PageRank.** We choose the PageRank-based centrality, defined as $\mathbf{V}_{PR}[i, i] = (1 - PR(i))^{-1}$ for each node $i \in \mathcal{V}$, where $PR(i)$ represents the PageRank score (Brin & Page, 1998). The PageRank score measures the probability that a random walke visits a specific node, making it a key metric for assessing node importance, especially in web search algorithms.

**Walk Count.** We define node centrality based on the number of walks of length $\ell$ starting from each node $i$, expressed as $\left( \mathbf{A}^{\ell} \mathbb{1} \right) [i]$, where $\mathbb{1} \in \mathbb{R}^N$ is a vector of ones.

Using this assumption, we employ node clustering to partition the node set into $C$ clusters, denoted by $\mathcal{P} = \cup_{1 \leq c \leq C} \mathcal{P}_c$. Each cluster $c \in \{1, \ldots, C\}$ is assigned a common exit layer $\ell_c \in \{0, \ldots, L\}$, determined using nodes excluded from the test set. Validation nodes are utilized for this purpose, as training nodes typically are usually well-predicted in all layers. *(i)* Centrality scores are computed for all nodes, considering metrics. *(ii)* Nodes are ranked based on their centrality scores *(iii)* These centrality scores are discretized into $C$ equal-sized buckets to facilitate the clustering process. *(iv)* The optimal exit layer for each cluster is determined by evaluating the classification accuracy on validation nodes within that cluster, i.e.,

$$\forall c \in \{1, \ldots, C\}, \quad \ell_c = \underset{\ell \in \{0, \ldots, L\}}{\arg \max} Acc \left\{ p_v^{(\ell)} \in \mathcal{V}_{val} \cap \mathcal{P}_c \right\}.$$

# 4 EXPERIMENTAL SETUP

## 4.1 DATASETS

We use thirteen widely used datasets in the GNN literature. We particularly used the citation networks Cora, CiteSeer, and PubMed Sen et al. (2008), the co-authorship networks CS (Shchur et al., 2018), the citation network between Computer Science arXiv papers ogbn-arxiv (Hu et al., 2020), the Amazon Computers and Amazon Photo networks (Shchur et al., 2018), the non-homophilous dataset genius (Lim & Benson, 2021), and the disassortative datasets Chameleon, Squirrel (Rozemberczki et al., 2021), and Cornell, Texas, Wisconsin from the WebKB dataset (Lim et al., 2021). More details and statistics about the used datasets can be found in Appendix C. For the Cora, CiteSeer, and Pubmed datasets, we used the provided train/validation/test splits. For the remaining datasets, we followed the framework in (Lim et al., 2021; Rozemberczki et al., 2021).

## 4.2 BASELINES

We compare our approach with architectures that combine all the hidden representations of nodes to form a final node representation used for prediction. For each baseline model, we vary the number of layers from 0 to 5, and we report in Table 3 the performance of the best number of layers with respect to the test set. *(i)* This includes *Jumping knowledge*, which combines the nodes representation of all layers using an aggregation layer, e.g., MaxPooling (JKMaxPool), Concatenation (JK-Concat), or LSTM-attention (JK-LSTM)Xu et al. (2018b). *(ii)* Residuals-GCNII which use an initial residual

Table 3: Classification accuracy ($\pm$ standard deviation) on different benchmark node classification datasets for the baselines based on the GCN backbone. The higher the accuracy (in %) the better the model. Highlighted are the **first**, second best results. OOM means *Out of memory*.

| Model | Cora | CiteSeer | CS | PubMed | genuis | ogbn-arxiv |
|---|---|---|---|---|---|---|
| JKNET-CAT (Xu et al., 2018b) | 79.52 (1.16) [2] | 69.69 (0.05) [1] | 91.23 (1.26) [1] | 77.63 (0.59) [2] | **81.46 (0.10) [2]** | 68.54 (0.57) [5] |
| JKNET-MAX (Xu et al., 2018b) | 75.67 (0.18) [1] | 70.12 (0.04) [1] | 90.55 (0.74) [1] | 75.10 (2.41) [2] | 80.13 (0.29) [1] | 56.66 (0.31) [5] |
| JKNET-LSTM (Xu et al., 2018b) | 78.95 (0.62) [0] | 65.83 (1.27) [0] | 90.17 (1.41) [2] | 77.73 (0.67) [0] | OOM | OOM |
| Residuals - GCNII (Chen et al., 2020) | 76.84 (0.20) [1] | 69.72 (0.06) [1] | 90.84 (1.35) [1] | 77.82 (0.44) [2] | **81.36 (1.13) [2]** | 61.48 (3.10) [4] |
| AdaGCN | 75.08 (0.27) | 69.58 (0.19) | 89.62 (0.51) | 76.40 (0.12) | 79.85 (0.00) | 22.06 (1.67) |
| GPR-GNN | 79.91 (0.43) [2] | 69.21 (0.81) [2] | 91.42 (1.12) [2] | 79.0 (0.39) [2] | 81.04 (0.41) [2] | 68.03 (0.23) [3] |
| GCN | 80.78 (0.72) [2] | 71.25 (0.72) [2] | **92.20 (0.00) [1]** | **79.32 (0.41) [2]** | 80.76 (1.05) [0] | 64.37 (0.43) [2] |
| ADMP-GCN | 81.22 (0.36) [5] | **71.33 (0.40) [2]** | 91.49 (0.66) [2] | 79.02 (0.21) [2] | 80.17 (0.79) [0] | 69.29 (0.74) [4] |
| ADMP-GCN w/ *Degree* | 81.03 (0.53) | 71.10 (0.50) | 91.26 (0.59) | 78.71 (0.39) | 80.73 (1.00) | 69.59 (0.28) |
| ADMP-GCN w/ *k-core* | **81.19 (0.40)** | 71.27 (0.53) | 91.29 (0.68) | 78.73 (0.41) | 80.73 (1.00) | 69.55 (0.33) |
| ADMP-GCN w/ *Walk Count* | 81.14 (0.41) | 71.14 (0.50) | 91.19 (0.64) | 78.64 (0.60) | 80.68 (0.97) | 69.55 (0.34) |
| ADMP-GCN w/ *PageRank* | 81.05 (0.46) | 71.0 (0.29) | 91.09 (0.99) | 78.69 (0.56) | 81.12 (1.41) | **69.60 (0.29)** |

connection and an identity mapping at each layer. he initial residual connection ensures that the final representation of each node retains at least a fraction of $\alpha$ from the input layer Chen et al. (2020). *(iii)* GPR-GCN which combines adaptive generalized PageRank (GPR) scheme with GNNs Chien et al. (2020). *(iv)* Ada-GCN, which proposes an RNN-like deep GNN architecture by incorporating AdaBoost to combine the layers Sun et al. (2019). To have a fair comparison, we trained ADMP-GNN as well as all the baselines under the same settings, and we fixed the maximum number of layers to $L = 5$.

### 4.3 IMPLEMENTATION DETAILS

We train all the models using the Adam optimizer Kingma & Ba (2014) and the same hyperparameters. The GNN hyperparameters in each dataset were performed using a Grid search on the classical GCN; we detail the values of these hyperparameters in Table 7 of Appendix D. To account for the impact of random initialization, each experiment was repeated 10 times, and the mean and standard deviation of the results were reported. The experiments have been run on both an NVIDIA A100 GPU and an RTX A6000 GPU.

## 5 EXPERIMENTAL RESULTS

Through extensive experiments on multiple datasets, we can better understand the scenarios in which ADMP-GCN proves to be effective. As observed in Tables 3, 4, 15, and 10, a comparison between ADMP-GCN and ADMP-GIN against their respective baselines GCN and GIN demonstrates consistently higher accuracy for most datasets. Regarding the centrality-based layer selection policy, it becomes clear that when graphs exhibit a wide range of local density and centrality among nodes, the centrality-based policy is particularly efficient. Most importantly, there is never a drop in accuracy observed with ADMP-GNN. However, beyond this observation, it is challenging to provide universal guidelines for selecting the most appropriate layer selection policy.

Table 4: Classification accuracy ($\pm$ standard deviation) on different benchmark node classification datasets for the baselines based on the GIN backbone. The higher the accuracy (in %), the better the model. Highlighted are the **first**, second best results. OOM means *Out of memory*.

| Model | Cora | CiteSeer | CS | PubMed | genuis | ogbn-arxiv |
|---|---|---|---|---|---|---|
| JKNET-CAT Xu et al. (2018b) | 77.94 (0.67) [2] | 64.82 (0.04) [1] | 89.26 (1.18) [1] | 75.89 (2.5) [2] | OOM | 60.23 (0.37) [1] |
| JKNET-MAX Xu et al. (2018b) | 77.47 (0.81) [1] | 64.96 (2.08) [2] | 87.4 (2.11) [0] | 76.21 (1.73) [0] | OOM | 51.84 (4.01) [0] |
| JKNET-LSTM Xu et al. (2018b) | 77.39 (1.39) [2] | 64.52 (2.02) [1] | 87.27 (3.66) [3] | 75.90 (1.63) [0] | OOM | OOM |
| GPR-GIN | 76.83 (1.22) [2] | 66.43 (1.15) [2] | 88.15 (1.53) [5] | **77.27 (0.87) [2]** | 80.82 (0.42) [1] | **63.05 (0.44) [2]** |
| GIN | 77.73 (0.99) [2] | 65.23 (1.45) [2] | 90.29 (0.99) [2] | 76.05 (1.14) [2] | 80.78 (1.03) [0] | 60.70 (0.15) [1] |
| ADMP-GIN | 78.07 (0.68) [2] | 65.41 (1.91) [2] | 90.82 (1.15) [1] | 76.46 (1.04) [4] | 80.47 (0.91) [0] | 60.85 (0.01) [1] |
| ADMP-GIN w/ *Degree* | 78.12 (0.70) | **66.82 (0.89)** | 90.70 (0.79) | 76.07 (1.27) | 81.68 (0.70) | 60.85 (0.01) |
| ADMP-GIN w/ *k-core* | 78.10 (0.64) | 66.36 (1.03) | **90.85 (0.93)** | 75.93 (1.13) | 81.48 (0.64) | 60.85 (0.01) |
| ADMP-GIN w/ *Walk Count* | **78.19 (0.68)** | 65.79 (0.81) | 90.77 (0.91) | 76.72 (0.76) | 81.21 (0.58) | 60.85 (0.01) |
| ADMP-GIN w/ *PageRank* | 77.78 (0.83) | 67.08 (0.51) | 90.72 (0.91) | 76.63 (0.87) | **81.89 (1.04)** | 60.85 (0.01) |

**Ablation Study.** The choice of using validation nodes rather than training nodes for learning the layer selection policy stems for the high prediction accuracy on training nodes across all layers.

This high accuracy leads to a significant distribution shift between the predictions for train nodes and those for test nodes. Conversely, validation nodes, which were not utilized during the training of the ADMP-GNN, present a more suitable option for learning the policy due to their unbiased predictions. We tested a variety of Deep Learning mechanisms to predict the best exit layer for each node. This included generalizing the policy by training neural networks to identify layers that accurately predict node outcomes or by framing the problem as an optimal stopping problem following the framework of Huré et al. (2021). However, these approaches require learning the policy on a set of nodes larger than the test set, which is impractical for node classification tasks where the training and validation node sets, available for policy learning, are typically smaller than the test set.

## 6  CONCLUSION

In this work, we have proposed ADMP-GNN, a novel adaption of message passing neural networks that enables any message passing neural network to make predictions for each node at every layer. Additionally, we have proposed a sequential training approach aimed at achieving results comparable to training multiple GNNs separately in a single-task setting. The empirical analysis of the results demonstrates the need for a node-specific depth in GNNs to better capture the unique characteristics and computational needs of each node. Determining the optimal number of message-passing layers for each node presents a significant challenge due to several factors. Graph structures vary widely in complexity and connectivity, influencing how information propagates through the network. Node features introduce further variability, impacting the effectiveness of message passing. However, through experiments, we identified instances where node centrality can help identify the optimal layer for each node. We heuristically learn a layer selection policy using a set of validation nodes. This policy is then generalized on the test nodes. Extensive experiments on multiple datasets demonstrate ADMP-GNN's effectiveness in improving the prediction accuracy of GNNs.

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

## A    NODE-SPECIFIC DEPTH ANALYSIS IN GRAPH NEURAL NETWORKS

We generate synthetic graphs of size $N = 5,000$. We select nodes belonging to sparse or dense regions in the original graph based on their core number. We consider only nodes with labels that are sufficiently present in both dense and sparse region. Last, we randomly select nodes, all by keeping the label distribution similar in both sparse and dense subgraphs.

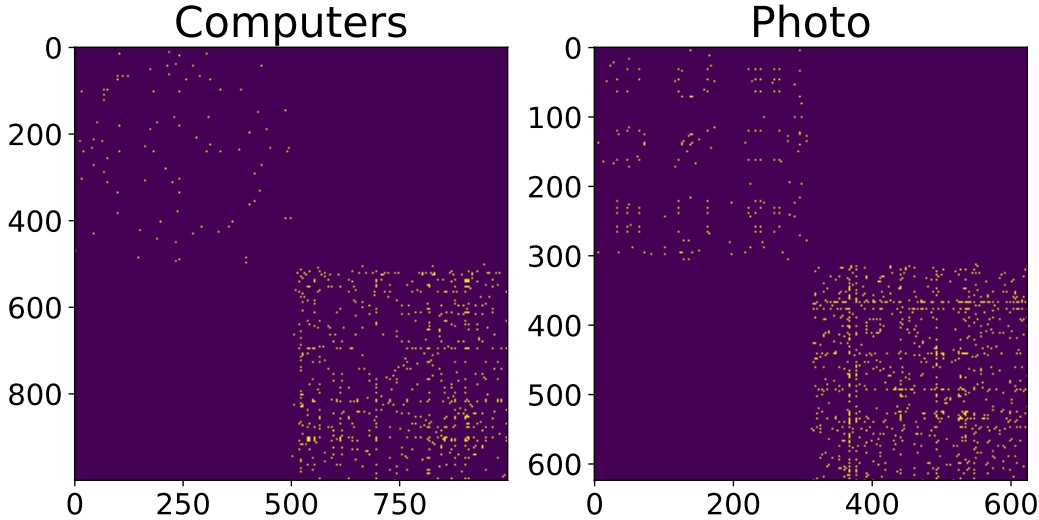

Figure 3: The adjacency matrix of the synthetic graphs extracted from the real graphs Computers and Photo.

## B    TIME COMPLEXITY

Table 5: The average time needed for each training setting for different datasets.

| Model | Cora | squirel | chamelon | Computers | Photo | Ogbn-arxiv |
|---|---|---|---|---|---|---|
| ADMP-GCN ALM | 14 | 36 | 15 | 51 | 26 | 266 |
| ADMP-GCN ST | 32 | 88 | 40 | 175 | 87 | 882 |

## C  DATASET STATISTICS

Table 6: Statistics of the node classification datasets used in our experiments.

| DATASET | #FEATURES | #NODES | #EDGES | #CLASSES | EDGE HOMOPHILY |
|---|---|---|---|---|---|
| CORA | 1,433 | 2,708 | 5,208 | 7 | 0.809 |
| CITESEER | 3,703 | 3,327 | 4,552 | 6 | 0.735 |
| PUBMED | 500 | 19,717 | 44,338 | 3 | 0.802 |
| CS | 6,805 | 18,333 | 81,894 | 15 | 0.808 |
| CHAMELEON | 2,325 | 2,277 | 62,792 | 5 | 0.231 |
| CORNELL | 1,703 | 183 | 557 | 5 | 0.132 |
| SQUIRREL | 2,089 | 5,201 | 396,846 | 5 | 0.222 |
| WISCONSIN | 1,703 | 251 | 916 | 5 | 0.206 |
| TEXAS | 1,703 | 183 | 574 | 5 | 0.111 |
| PHOTO | 745 | 7,650 | 238,162 | 8 | 0.827 |
| OGBN-ARXIV | 128 | 169,343 | 2,315,598 | 40 | 0.654 |
| COMPUTERS | 767 | 13752 | 491,722 | 10 | 0.777 |

## D  HYPERPARAMETER CONFIGURATIONS

Table 7: Hyperparameters used in our experiments.

| DATASET | HIDDEN SIZE | LEARNING RATE | DROPOUT PROBABILITY |
|---|---|---|---|
| CORA | 64 | 0.01 | 0.8 |
| CITESEER | 64 | 0.01 | 0.4 |
| PUBMED | 64 | 0.01 | 0.2 |
| CS | 512 | 0.01 | 0.4 |
| ARXIV-YEAR | 512 | 0.01 | 0.2 |
| CHAMELEON | 512 | 0.01 | 0.2 |
| CORNELL | 512 | 0.01 | 0.2 |
| DEEZER-EUROPE | 512 | 0.01 | 0.2 |
| SQUIRREL | 512 | 0.01 | 0.2 |
| WISCONSIN | 512 | 0.01 | 0.2 |
| TEXAS | 512 | 0.01 | 0.2 |
| PHOTO | 512 | 0.01 | 0.6 |
| OGBN-ARXIV | 512 | 0.01 | 0.5 |
| COMPUTERS | 512 | 0.01 | 0.2 |
| PHYSICS | 512 | 0.01 | 0.4 |
| PENN94 | 64 | 0.01 | 0.2 |

# E    ADDITIONAL EXPERIMENTS ON GCN

Table 8: Comparison of three GCN training settings: *GCN* – trained separately for each layer; *ADMP-GCN ALM* trained across all layers simultaneously using backpropagation on the summed loss; *ADMP-GCN ST* trained using Dynamic Programming. **(*)** denotes single-task training.

| #layers | model | Dataset | | | | | | |
|---|---|---|---|---|---|---|---|---|
| | | Photo | Computers | chamelon | Cornell | Wisconsin | Texas | squirrel |
| 0 | GCN(*) | 70.18 (3.39) | 56.83 (2.92) | 33.55 (0.00) | 40.54 (0.00) | 70.59 (0.00) | 64.86 (0.00) | 26.42 (0.00) |
| | ADMP-GCN *ALM* | 68.23 (3.11) | 56.61 (2.07) | 30.70 (0.00) | 40.54 (0.00) | 52.94 (0.00) | 64.86 (0.00) | 22.60 (0.06) |
| | ADMP-GCN *ST* | 70.56 (3.33) | 57.94 (3.62) | 33.55 (0.00) | 40.54 (0.00) | **70.59 (0.00)** | **64.86 (0.00)** | 26.42 (0.00) |
| 1 | GCN(*) | 65.86 (5.18) | 59.62 (3.40) | 38.16 (0.00) | 40.54 (0.00) | 56.86 (0.00) | 64.86 (0.00) | 21.23 (0.00) |
| | ADMP-GCN *ALM* | 64.49 (6.95) | 57.60 (6.63) | 27.41 (0.00) | 40.54 (0.00) | 54.90 (0.00) | 64.86 (0.00) | 19.31 (0.00) |
| | ADMP-GCN *ST* | 64.03 (3.63) | 56.61 (5.44) | 38.16 (0.00) | 40.54 (0.00) | 56.86 (0.00) | 64.86 (0.00) | 21.23 (0.00) |
| 2 | GCN(*) | 82.32 (2.97) | 69.05 (2.60) | **58.73 (0.96)** | 54.59 (2.91) | 57.06 (1.85) | 62.16 (1.21) | 33.40 (0.68) |
| | ADMP-GCN *ALM* | 48.10 (6.28) | 34.06 (10.87) | 40.39 (0.95) | 44.32 (1.32) | 56.86 (1.52) | 62.16 (0.00) | 21.73 (0.28) |
| | ADMP-GCN *ST* | 85.80 (0.43) | 68.22 (4.53) | 58.77 (1.08) | 55.95 (4.02) | 56.86 (1.52) | 61.89 (1.46) | 33.57 (0.46) |
| 3 | GCN(*) | 86.51 (2.33) | **75.32 (4.18)** | 58.40 (2.46) | 45.95 (3.42) | 43.33 (2.23) | 44.59 (4.05) | 36.29 (0.73) |
| | ADMP-GCN *ALM* | 79.78 (4.17) | 53.40 (9.51) | 49.67 (1.03) | 38.92 (5.43) | 46.47 (2.16) | 50.00 (3.25) | 29.65 (2.22) |
| | ADMP-GCN *ST* | 85.82 (1.76) | 73.69 (3.67) | 50.79 (1.02) | 48.38 (3.07) | 47.25 (2.23) | 57.30 (5.38) | 32.95 (0.45) |
| 4 | GCN(*) | 75.36 (12.68) | 61.77 (20.27) | 51.12 (7.56) | 47.03 (2.16) | 51.57 (2.91) | 48.38 (3.51) | **37.73 (0.80)** |
| | ADMP-GCN *ALM* | 82.37 (4.86) | 62.61 (5.71) | 49.63 (1.90) | 44.59 (4.23) | 48.04 (2.67) | 52.43 (4.86) | 28.99 (1.40) |
| | ADMP-GCN *ST* | **87.08 (0.77)** | 72.35 (5.44) | 53.86 (2.44) | 51.35 (3.20) | 50.39 (1.76) | 57.84 (5.01) | 34.74 (0.48) |
| 5 | GCN(*) | 78.49 (7.79) | 55.63 (14.13) | 50.07 (1.73) | 46.76 (4.53) | 45.88 (2.18) | 53.51 (6.02) | 36.52 (1.01) |
| | ADMP-GCN *ALM* | 82.57 (5.33) | 63.05 (9.21) | 48.36 (1.86) | 48.92 (3.72) | 46.67 (4.71) | 49.46 (5.93) | 29.60 (1.70) |
| | ADMP-GCN *ST* | 86.37 (0.85) | 74.53 (3.65) | 53.86 (1.39) | 52.70 (3.02) | 50.59 (1.47) | 56.49 (3.07) | 32.11 (1.14) |

Table 9: The first two rows of the table present the highest accuracy (± standard deviation) for *GCN* and *ADMP-GCN ST*, with the corresponding layer where this accuracy is achieved indicated in brackets [.]. The final row reports the *Oracle accuracy* for *ADMP-GCN ST*.

| model | Dataset | | | | | | |
|---|---|---|---|---|---|---|---|
| | Photo | Computers | chamelon | Cornell | Wisconsin | Texas | squirrel |
| GCN | 86.51 (2.33) [3] | 75.32 (4.18) [3] | 58.73 (0.96) [2] | 54.59 (2.91) [2] | 70.59 (0.00) [0] | 64.86 (0.00) [0] | 37.73 (0.80) [4] |
| ADMP-GCN DT | 87.08 (0.77) [4] | 74.53 (3.65) [5] | 58.77 (1.08) [2] | 55.95 (4.02) [2] | 70.59 (0.00) [0] | 64.86 (0.00) [0] | 34.74 (0.48) [4] |
| ADMP-GCN ST - Oracle | **96.14 (0.81)** | **90.15 (0.97)** | **79.5 (1.31)** | **74.59 (3.24)** | **80.39 (0.00)** | **77.84 (2.36)** | **69.54 (0.65)** |

Table 10: Classification accuracy (± standard deviation) on different benchmark node classification datasets for the baselines based on the GCN backbone. The higher the accuracy (in %) the better the model. Highlighted are the **first**, second best results. OOM means *Out of memory*.

| Model | Photo | Computers | chamelon | Cornell | Wisconsin | Texas | squirrel |
|---|---|---|---|---|---|---|---|
| JKNET-CAT Xu et al. (2018b) | 87.92 (1.98) [2] | 74.68 (6.92) [3] | 56.89 (1.77) [2] | 46.22 (5.73) [5] | 72.55 (0.00) [0] | 64.86 (0.00) [0] | 41.26 (0.88) [2] |
| JKNET-MAX Xu et al. (2018b) | 88.02 (2.21) [2] | **77.97 (2.57) [0]** | 57.70 (2.79) [2] | 45.95 (5.41) [5] | 62.75 (2.63) [1] | 68.65 (4.05) [5] | **41.36 (0.59) [2]** |
| JKNET-LSTM Xu et al. (2018b) | 87.74 (1.94) [0] | 77.13 (2.80) [1] | 53.07 (4.47) [2] | 43.78 (3.78) [2] | 62.94 (2.23) [1] | 64.86 (0.00) [0] | 41.32 (0.58) [1] |
| Residuals - GCNII Chen et al. (2020) | 86.99 (2.47) [2] | 62.83 (19.61) [2] | **61.03 (2.07) [2]** | 55.14 (8.48) [5] | 70.59 (0.00) [0] | 64.86 (0.00) [0] | 35.90 (1.25) [5] |
| AdaGCN | 64.84 (0.89) | 58.68 (1.31) | 47.37 (0.48) | **70.00 (4.75)** | **75.88 (1.76)** | **72.43 (1.62)** | 29.15 (0.77) |
| GPR-GNN | **89.16 (2.04) [2]** | 77.43 (3.32) [2] | 63.29 (0.91) [2] | 62.70 (4.65) [2] | 58.82 (1.24) [2] | 59.19 (3.72) [2] | 38.52 (1.06) [4] |
| GCN | 86.51 (2.33) [3] | 75.32 (4.18) [3] | 58.73 (0.96) [2] | 54.59 (2.91) [2] | 70.59 (0.00) [0] | 64.86 (0.00) [0] | 37.73 (0.80) [4] |
| ADMP-GCN | 87.08 (0.77) [4] | 74.53 (3.65) [5] | 58.77 (1.08) [2] | 55.95 (4.02) [2] | 70.59 (0.00) [0] | 64.86 (0.00) [0] | 34.74 (0.48) [4] |
| ADMP-GCN w/ *Degree* | 88.15 (1.52) | 75.53 (2.09) | 58.75 (0.91) | 44.32 (4.22) | 63.92 (1.30) | 56.49 (1.89) | 35.07 (1.13) |
| ADMP-GCN w/ *k-core* | 88.31 (1.31) | 75.57 (1.88) | 58.97 (1.09) | 48.11 (4.65) | 68.82 (1.37) | 60.81 (3.25) | 34.46 (0.70) |
| ADMP-GCN w/ *Walk Count* | 88.42 (1.51) | 76.14 (2.08) | 58.29 (1.22) | 51.89 (4.32) | 67.45 (1.80) | 68.65 (2.16) | 34.39 (0.89) |
| ADMP-GCN w/ *PageRank* | 88.21 (1.45) | 75.50 (2.23) | 58.44 (1.23) | 55.14 (4.86) | 65.29 (2.16) | 60.00 (3.15) | 34.68 (0.66) |

# F  EXPERIMENTS ON GIN

Table 11: Comparison of three GIN training settings: *GIN* trained separately for each layer; *ADMP-GIN ALM* trained across all layers simultaneously using backpropagation on the summed loss; *ADMP-GIN ST* trained using Dynamic Programming. **(\*)** denotes single-task training.

| #layers | model | Dataset | | | | | |
|---|---|---|---|---|---|---|---|
| | | Cora | CiteSeer | CS | PubMed | Genius | ogbn-arxiv |
| 0 | GIN(\*) | 56.40 (0.06) | 57.14 (0.09) | 87.17 (1.41) | 72.49 (0.08) | 80.78 (1.03) | 48.87 (0.04) |
| | ADMP-GIN *ALM* | 58.00 (0.00) | 61.50 (0.00) | 86.36 (0.78) | 73.20 (0.00) | 79.95 (0.10) | 36.49 (0.19) |
| | ADMP-GIN *ST* | 56.38 (0.04) | 57.17 (0.08) | 87.41 (0.95) | 72.47 (0.14) | 80.47 (0.91) | 48.87 (0.04) |
| 1 | GIN(\*) | 75.17 (0.09) | 64.79 (0.03) | 90.29 (0.99) | 74.97 (0.11) | 78.42 (4.95) | 60.9 (0.15) |
| | ADMP-GIN *ALM* | 74.50 (0.00) | 66.50 (0.00) | 88.66 (1.18) | 75.40 (0.00) | 72.07 (17.44) | 59.43 (0.73) |
| | ADMP-GIN *ST* | 75.07 (0.06) | 64.80 (0.00) | **90.82 (1.15)** | 75.00 (0.00) | 77.01 (12.06) | **60.85 (0.01)** |
| 2 | GIN(\*) | **77.73 (0.99)** | **65.23 (1.45)** | 87.93 (0.71) | 76.05 (1.14) | 78.89 (0.48) | 16.23 (9.60) |
| | ADMP-GIN *ALM* | 63.53 (4.80) | 60.68 (2.09) | 12.90 (8.05) | 72.20 (4.11) | 79.60 (0.51) | 47.63 (3.20) |
| | ADMP-GIN *ST* | 78.07 (0.68) | 65.41 (1.91) | 87.47 (2.18) | 76.04 (0.88) | 72.99 (13.48) | 10.13 (8.00) |
| 3 | GIN(\*) | 74.40 (1.14) | 60.81 (2.20) | 82.13 (2.24) | 74.97 (1.69) | 52.22 (28.47) | 6.00 (0.17) |
| | ADMP-GIN *ALM* | 66.99 (2.85) | 59.57 (2.57) | 17.69 (12.02) | 74.15 (2.00) | 68.00 (23.98) | 27.51 (16.25) |
| | ADMP-GIN *ST* | 76.28 (1.11) | 65.13 (1.00) | 83.60 (3.72) | **76.21 (1.75)** | 80.04 (0.09) | 13.74 (9.66) |
| 4 | GIN(\*) | 67.68 (4.07) | 57.54 (2.92) | 47.37 (17.10) | 73.62 (1.63) | **80.05 (0.10)** | 6.07 (0.21) |
| | ADMP-GIN *ALM* | 68.78 (4.30) | 60.12 (2.36) | 21.07 (14.15) | 74.20 (2.57) | 79.36 (1.18) | 15.23 (11.85) |
| | ADMP-GIN *ST* | 74.94 (1.58) | 65.21 (1.54) | 81.33 (2.15) | 76.46 (1.04) | 80.04 (0.09) | 16.02 (10.78) |
| 5 | GIN(\*) | 32.48 (10.47) | 54.76 (2.32) | 18.56 (11.32) | 67.98 (5.80) | 80.05 (0.10) | 6.19 (0.39) |
| | ADMP-GIN *ALM* | 67.46 (4.34) | 59.74 (1.95) | 29.16 (13.84) | 73.94 (2.42) | 79.99 (0.10) | 14.61 (12.11) |
| | ADMP-GIN *ST* | 71.34 (2.09) | 63.89 (1.39) | 79.10 (1.84) | 75.75 (1.09) | 79.57 (1.40) | 15.72 (12.13) |

Table 12: Comparison of three GIN training settings: *GIN* trained separately for each layer; *ADMP-GIN ALM* trained across all layers simultaneously using backpropagation on the summed loss; *ADMP-GIN ST* trained using Dynamic Programming.**(\*)** denotes single-task training.

| #layers | model | Dataset | | | | | | |
|---|---|---|---|---|---|---|---|---|
| | | Photo | Computers | chameleon | Cornell | Wisconsin | Texas | squirrel |
| 0 | GIN(\*) | 70.19 (2.91) | 56.83 (3.89) | 33.55 (0.00) | 40.54 (0.00) | 70.59 (0.00) | 64.86 (0.00) | 26.42 (0.00) |
| | ADMP-GIN *ALM* | 69.16 (3.27) | 56.27 (3.0) | 30.7 (0.00) | 40.54 (0.00) | 52.94 (0.00) | 64.86 (0.00) | 22.62 (0.05) |
| | ADMP-GIN *ST* | 68.68 (3.36) | 56.71 (2.77) | 33.55 (0.00) | 40.54 (0.00) | 70.59 (0.00) | 64.86 (0.00) | **26.42 (0.00)** |
| 1 | GIN(\*) | 82.32 (2.03) | 70.9 (2.64) | **61.03 (0.10)** | 40.54 (0.00) | **56.86 (0.00)** | 64.86 (0.00) | 47.65 (0.27) |
| | ADMP-GIN *ALM* | 78.79 (2.84) | 65.28 (2.17) | 56.75 (0.09) | 40.54 (0.00) | 54.9 (0.00) | 64.86 (0.00) | 44.12 (0.39) |
| | ADMP-GIN *ST* | 83.35 (1.67) | **71.88 (3.80)** | 60.96 (0.00) | 40.54 (0.00) | 56.86 (0.00) | 64.86 (0.00) | 47.65 (0.00) |
| 2 | GIN(\*) | **83.86 (2.19)** | 63.39 (10.3) | 63.57 (1.01) | **61.62 (2.65)** | 54.71 (2.7) | 64.86 (2.96) | 19.84 (1.59) |
| | ADMP-GIN *ALM* | 25.83 (9.5) | 13.04 (8.6) | 22.37 (0.00) | 40.0 (3.97) | 49.61 (3.51) | 64.32 (2.91) | 20.11 (0.85) |
| | ADMP-GIN *ST* | 68.44 (23.78) | 42.65 (17.47) | 62.3 (2.18) | 59.46 (2.42) | 53.33 (2.6) | 64.32 (1.62) | 19.31 (0.00) |
| 3 | GIN(\*) | 28.63 (13.64) | 19.9 (9.2) | 26.29 (1.94) | 35.68 (2.65) | 48.04 (4.04) | 60.54 (5.82) | 24.84 (2.06) |
| | ADMP-GIN *ALM* | 17.94 (6.7) | 23.04 (13.93) | 26.29 (1.31) | 44.05 (4.53) | 50.2 (5.13) | 64.59 (7.69) | 19.95 (2.4) |
| | ADMP-GIN *ST* | 71.32 (19.8) | 54.4 (16.24) | 48.84 (2.76) | 43.51 (3.51) | 56.27 (5.26) | 64.59 (14.58) | 26.03 (0.52) |
| 4 | GIN(\*) | 14.43 (4.93) | 12.51 (8.6) | 29.63 (2.48) | 43.78 (4.15) | 51.57 (1.53) | 66.76 (4.69) | 20.47 (1.26) |
| | ADMP-GIN *ALM* | 11.81 (6.16) | 9.01 (9.74) | 21.01 (2.16) | 45.14 (4.02) | 50.78 (3.22) | 64.05 (3.83) | 19.18 (1.57) |
| | ADMP-GIN *ST* | 68.49 (16.64) | 52.61 (19.82) | 44.54 (4.16) | 45.14 (4.84) | 50.59 (3.14) | 68.92 (5.7) | 26.59 (1.36) |
| 5 | GIN(\*) | 14.22 (5.42) | 12.55 (7.97) | 26.47 (3.22) | 43.51 (4.43) | 50.2 (6.09) | 66.22 (1.81) | 20.6 (1.25) |
| | ADMP-GIN *ALM* | 15.84 (4.93) | 9.81 (7.59) | 28.82 (5.47) | 42.97 (5.33) | 49.41 (4.09) | 65.14 (5.05) | 19.77 (1.52) |
| | ADMP-GIN *ST* | 68.48 (13.8) | 58.96 (9.83) | 43.31 (3.01) | 44.05 (4.53) | 45.69 (5.19) | **67.57 (9.44)** | 29.74 (2.46) |

Table 13: The first two rows of the table present the highest accuracy ($\pm$ standard deviation) for *GCN* and *ADMP-GCN ST*, with the corresponding layer where this accuracy is achieved indicated in brackets [.]. The final row reports the *Oracle accuracy* for *ADMP-GIN ST*.

| model | Dataset | | | | | |
|---|---|---|---|---|---|---|
| | Cora | CiteSeer | CS | PubMed | Genius | ogbn-arxiv |
| GIN | 77.73 (0.99) [2] | 65.23 (1.45) [2] | 90.29 (0.99) [1] | 76.05 (1.14) [2] | 80.78 (1.03) [0] | 60.9 (0.15) [1] |
| ADMP-GIN *ST* | 78.07 (0.68) [2] | 65.41 (1.91) [2] | 90.82 (1.15) [1] | 76.46 (1.04) [4] | 80.47 (0.91) [0] | 60.85 (0.01) [1] |
| ADMP-GIN *ST* - Oracle | **90.76 (0.27)** | **81.87 (0.63)** | **97.64 (0.18)** | **92.73 (0.54)** | **92.07 (6.13)** | **71.23 (2.69)** |

Table 14: The first two rows of the table present the highest accuracy (± standard deviation) for *GCN* and *ADMP-GCN ST*, with the corresponding layer where this accuracy is achieved indicated in brackets [.]. The final row reports the *Oracle accuracy* for *ADMP-GIN ST*.

| model | Dataset | | | | | | |
|---|---|---|---|---|---|---|---|
| | Photo | Computers | chamelon | Cornell | Wisconsin | Texas | squirrel |
| GIN | 83.86 (2.19) [2] | 70.9 (2.64) [1] | 63.57 (1.01) [2] | 61.62 (2.65) [2] | 70.59 (0) [0] | 66.76 (4.69) [4] | 47.65 (0.27) [1] |
| ADMP-GIN *ST* | 83.35 (1.67) [1] | 71.88 (3.8) [1] | 62.3 (2.18) [2] | 59.46 (2.42) [2] | 70.59 (0) [0] | 68.92 (5.7) [4] | 47.65 (0.0) [1] |
| ADMP-GIN *ST* - Oracle | **95.92 (1.03)** | **90.4 (2.66)** | **86.73 (0.9)** | **73.24 (4.75)** | **80.78 (1.18)** | **84.05 (3.07)** | **78.41 (0.89)** |

Table 15: Classification accuracy (± standard deviation) on different benchmark node classification datasets for the baselines based on the GIN backbone. The higher the accuracy (in %) the better the model. Highlighted are the **first**, second best results. OOM means *Out of memory*.

| Model | Photo | Computers | chamelon | Cornell | Wisconsin | Texas | squirrel |
|---|---|---|---|---|---|---|---|
| JKNET-MAX Xu et al. (2018b) | 82.48 (2.95) [1] | 69.62 (3.57) [1] | 59.61 (2.24) [2] | 47.84 (4.84) [4] | 57.84 (2.19) [0] | 66.49 (2.16) [5] | 23.19 (3.55) [2] |
| JKNET-LSTM Xu et al. (2018b) | **87.02 (1.42) [2]** | **76.50 (1.44) [2]** | 62.61 (1.39) [2] | 50.54 (3.64) [2] | 57.84 (3.64) [4] | 70.81 (3.15) [1] | 26.34 (4.31) [0] |
| GPR-GIN | 85.82 (1.02) [2] | 76.01 (3.2) [3] | **67.26 (0.5) [2]** | **62.97 (2.97) [5]** | 68.04 (3.62) [4] | 73.78 (2.72) [5] | 46.78 (1.8) [2] |
| GIN | 83.86 (2.19) [2] | 70.9 (2.64) [1] | 63.57 (1.01) [2] | 61.62 (2.65) [2] | 70.59 (0.00) [0] | 66.76 (4.69) [4] | 47.65 (0.27) [1] |
| ADMP-GIN | 83.35 (1.67) [1] | 71.88 (3.8) [1] | 62.3 (2.18) [2] | 59.46 (2.42) [2] | 70.59 (0.00) [0] | 68.92 (5.7) [4] | **47.65 (0.00) [1]** |
| ADMP-GIN w/ *Degree* | 84.15 (1.17) | 74.55 (2.89) | 64.43 (1.47) | 46.49 (4.65) | 66.47 (2.83) | 69.46 (3.21) | 47.65 (0.00) |
| ADMP-GIN w/ *k-core* | 84.08 (1.09) | 74.58 (3.56) | 64.65 (1.45) | 46.76 (4.84) | **70.78 (4.68)** | 71.89 (4.22) | 47.65 (0.00) |
| ADMP-GIN w/ *Walk Count* | 84.03 (1.31) | 73.85 (3.70) | 63.38 (1.53) | 58.11 (6.07) | 63.14 (3.90) | **74.86 (2.97)** | 47.65 (0.00) |
| ADMP-GIN w/ *PageRank* | 84.36 (1.32) | 74.27 (2.97) | 63.31 (1.1) | 55.41 (3.68) | 65.49 (3.3) | 70.81 (3.78) | 47.65 (0.00) |

