# OpenReview forum: "ADMP-GNN: Adaptive Depth Message Passing GNN"
_ICLR.cc/2025/Conference — ICLR 2025 Conference Withdrawn Submission_

### Official Review · Reviewer_qkSL · 2024-10-15

**Soundness:** 3
**Presentation:** 3
**Contribution:** 3
**Rating:** 6
**Confidence:** 3

**Summary:**

This paper proposed Adaptive Depth Message Passing GNN (ADMP-GNN), which learns a preset-max-depth GNN, and outputs predictions per layer. The authors also proposed training tricks and layer selection heuristics during test time. The network is efficient and shown to be effective on some real-world datasets.

**Strengths:**

- Overall, the writing is pretty good and clear.
- The motivation about adaptive layer for each node is convincing.
- The synthetic dataset experiment is pretty well designed.
- The architecture is efficient and has the same complexity as a normal GNN.
- The empirical experiments are promising and show the model-agnostic property of the proposed method. The ablations are extensive.

**Weaknesses:**

- Some related works are missing. The authors claim none work has been done for adaptive layer GNNs, which is not true. There are [1] [2] [3] [4], which may not have exactly the same method, but they also share the idea of adaptive message passing for nodes.
- Similarly, these works should also be compared in the experiments.

[1] Spinelli, Indro, Simone Scardapane, and Aurelio Uncini. "Adaptive propagation graph convolutional network." IEEE Transactions on Neural Networks and Learning Systems 32.10 (2020): 4755-4760.
[2] Finkelshtein, Ben, et al. "Cooperative graph neural networks." arXiv preprint arXiv:2310.01267 (2023).
[3] Faber, Lukas, and Roger Wattenhofer. "GwAC: GNNs with Asynchronous Communication." Learning on Graphs Conference. PMLR, 2024.
[4] Errica, Federico, et al. "Adaptive Message Passing: A General Framework to Mitigate Oversmoothing, Oversquashing, and Underreaching." arXiv preprint arXiv:2312.16560 (2023).

**Questions:**

- When executing the experiments for figure 1, are the L different GNNs trained with the same dynamics? (batch size, epochs, learning rate ...) Because deeper models might be harder to train. See also [5], these authors questioned the oversmoothing issue, and they claim it's just trainability issue for deep networks.
- In figure 1, I can understand the green curves have an upside-down V shape, too few layers lead to under-reaching, and too many layers lead to over-smoothing. But why are the blue curves V shape? It does not make sense to me, why it drops first and adding more layers is better again.
- In line 193-194, is there a motivation why no hidden representations h is included in the exit function?
- In line 236, the authors mentioned about the conflicts in gradients. Is there any proof or related work mentioning this?
- I wonder if stochastic depth [6] training also helps, which can be a candidate to your sequential training trick.
- Is this work directly applicable to graph property prediction, like, just by adding a pooling layer? If yes, why no experiments on graph prediction tasks? If not, why?
- In line 313, 316, what is ADMP-GNN PT?
- Some experiment results seem pretty bad, for example, in table 1, CS column, 2 layers, ADMP-GNN ALM is much worse than GCN(*). Why? Because of the gradient conflicts?

[5] Peng, Jie, Runlin Lei, and Zhewei Wei. "Beyond Over-smoothing: Uncovering the Trainability Challenges in Deep Graph Neural Networks." arXiv preprint arXiv:2408.03669 (2024).
[6] Huang, Gao, et al. "Deep networks with stochastic depth." Computer Vision–ECCV 2016: 14th European Conference, Amsterdam, The Netherlands, October 11–14, 2016, Proceedings, Part IV 14. Springer International Publishing, 2016.

---

### Official Review · Reviewer_3SQN · 2024-11-01

**Soundness:** 3
**Presentation:** 3
**Contribution:** 3
**Rating:** 3
**Confidence:** 3

**Summary:**

The key idea of this work is to use varying numbers of message-passing steps to aggregate information uniquely for each node, allowing different nodes to utilize different aggregation and propagation layers. After a certain number of layers, the model first calculates the loss for these nodes, performs backpropagation, and then freezes their gradients to avoid conflicts in subsequent aggregation layers. The paper proposes an adaptive framework to determine the optimal number of layers for each node.

**Strengths:**

1. The idea is novel to me. However, since I haven’t worked on GNNs since 2022, I’m unable to provide an accurate assessment of its novelty.

2. The paper proposes a sequential training approach, which helps reduce computational complexity.

3. To improve generalization on test nodes, the paper introduces several heuristic metrics—k-core, PageRank, and Walk Count—to determine the optimal number of aggregation layers.

**Weaknesses:**

1. The improvement is somewhat limited on certain datasets, such as Cora and Citeseer.

2. The paper is under 9 pages, and, in my view, it feels somewhat incomplete.

**Questions:**

N/A

---

### Official Review · Reviewer_onjg · 2024-11-03

**Soundness:** 2
**Presentation:** 3
**Contribution:** 2
**Rating:** 3
**Confidence:** 4

**Summary:**

This paper explores using node-wise adaptive depth aggregation for GNN. To this end, a progressive training strategy is proposed, and the proposed method is evaluated on several benchmark datasets.

**Strengths:**

The paper is well-written and the adaptive depth aggregation is well-motivated.

**Weaknesses:**

It is understandable that different nodes may benefit from different depths of aggregation, and the results in Table 2 show that the maximum test accuracy by using the aggregation results from different layers is much better than what is achieved by using the aggregation results from a single layer. However, the main challenge is determining the optimal depth for each node. The heuristic method proposed in this paper utilizes clustering and then uses the validation accuracy for each cluster to determine the optimal depth for the whole cluster that can easily not be optimal, e.g., maybe, in some clusters, the validation nodes are not too small and not representative enough. Indeed, the accuracy of the proposed method is not much better than the GCN baseline.

**Questions:**

How well does the centrality-based layer selection policy algin with the one given in the oracle accuracy?

---

### Official Review · Reviewer_84Vi · 2024-11-03

**Soundness:** 2
**Presentation:** 2
**Contribution:** 1
**Rating:** 3
**Confidence:** 4

**Summary:**

This paper proposes an adaptive message-passing scheme that allows the updating of nodes in an independent way with respect to the number of message-passing steps (layers). The authors propose several ways to train the network using this mechanism, and several test time usages with different encodings.  The paper concludes with several experiments to demonstrate the performance of ADMP-GNN.

**Strengths:**

Strengths:
1. As said by the authors, it is sensible that adaptivity is important (in NNs and also in GNNs).

2. The adaptivity mechanism itself (not the method as a whole) seems simple (which is good).

**Weaknesses:**

Weaknesses:

1. The paper lacks discussions on highly relevant papers that suggest adaptivity in GNNs, from the message-passing (most important for this paper), and other mechanisms like the activation function and the normalization layer. Please see references R1-R5 below.

2. Regarding W1, the papers R1 and R2 seem to do the same basic idea as here (even if the implementation is a bit different), so it does not seem novel, and it is also not discussed in this paper.

3. Adaptivity in message passing can also be obtained with graph transformers/graph attention layers because they can adaptively choose which nodes should be discussed with which, but the authors did not discuss them in the paper or within the experiments.  Similarly, the method should be compared with rewiring methods.

4. It is not clear what the main benefit of this method is beyond adaptivity (which is important). For example, there is no discussion on which inherent problems in GNNs it can solve, like over-smoothing or over-squashing, and there is also no discussion on the expressiveness of this method.

5. Regarding the design of the adaptive mechanism (section 3.2), it is interesting to know what the network actually learns to adapt to and how. However, this was not shown in the paper, so it lacks insights.

6. The time complexity of the method is quite high. Although it is discussed by the authors, they do not properly compare the required runtimes with other methods, and they should also compare their performance with other methods that require more time, such as graph transformers and subgraph methods.

7. I am a bit concerned about section 3.5; it shows heuristics to use the method on test nodes, however these proposed heuristics and encodings were also used in other methods and were shown to be quite strong, for example, the 'Walk Count' was used in the well-known RWSE encoding (see R6 for reference), and I think that it is hard to disentangle the results obtained by these approaches from the actual method proposed here, which is the adaptive message passing mechanism.

**Regarding experiments, I have several concerns:**

8. First, it is only conducted on up to 5 layers. However, most GNNs perform quite well in this regime (e.g., see R6). It is more interesting to know how this method performs when there are many layers, like 64,128,256, etc.

9. It is not clear which splits were used. While the authors do say they use 10 seeds, the splitting of the data is not clear and does not seem to be in line with known splits of the datasets used here (e.g., as in R7).

10. The results are poorly compared with relevant methods, as previously discussed, and this makes it hard to quantify the contribution of the proposed method properly. Also, the uncertainty on which splits were used in the experiments makes it harder to compare with known results from previous papers.

11. The datasets used here were shown to suffer from multiple problems (see R8, R9), and therefore, showing results on additional and more up-to-date datasets is required. I would strongly suggest trying to benchmark your method on those shown in R9 and additional tasks like graph classification or regression.

12. The ablation study is discussed in the main paper, but the results are missing.

[R1] Adaptive Message Passing: A General Framework to Mitigate Oversmoothing, Oversquashing, and Underreaching

[R2] Cooperative Graph Neural Networks

[R3]  Improving Expressivity of GNNs with Subgraph-specific Factor Embedded Normalization

[R4] GRANOLA: Adaptive Normalization for Graph Neural Networks

[R5] DiGRAF: Diffeomorphic Graph-Adaptive Activation Function

[R6] A Survey on Oversmoothing in Graph Neural Networks

[R7] Geom-GCN: Geometric Graph Convolutional Networks

[R8] Pitfalls of Graph Neural Network Evaluation

[R9] A critical look at the evaluation of GNNs under heterophily: Are we really making progress?

**Questions:**

Please see the discussed weaknesses.

---

### Note · Authors · 2024-12-04

**Comment:**

We would very much like to thank the reviewers for their insightful reviews and the AC for taking the time to handle our paper. Unfortunately, we were unable to produce a comprehensive, convincing rebuttal during the discussion period and would therefore like to withdraw the paper to not take anymore of your time. We will certainly take the valuable feedback provided into account in future versions of this work. Thank you very much again.

**Withdrawal Confirmation:**

I have read and agree with the venue's withdrawal policy on behalf of myself and my co-authors.